# Long Stroke Design of Piezoelectric Walking Actuator for Wafer Probe Station

**DOI:** 10.3390/mi13030412

**Published:** 2022-03-05

**Authors:** Cheng Yang, Yin Wang, Wei Fan

**Affiliations:** 1Institute of Manufacturing Engineering, Huaqiao University, Xiamen 361021, China; 17013080060@stu.hqu.edu.cn; 2College of Mechanical Engineering and Automation, Huaqiao University, Xiamen 361021, China; fanwei@hqu.edu.cn

**Keywords:** wafer probe station, inchworm mechanism, long-stroke, piezoelectric actuator

## Abstract

In order to develop a high-resolution piezoelectric walking actuator with a long stroke for the wafer probe station, this work presents a design of a piezoelectric walking actuator with two auxiliary clamping feet elastically attached to major clamping feet. Its construction was introduced and its operating principle was analyzed. Structure design details were discussed and a prototype was proposed. The prototype was fabricated and tested. The experimental results show that the proposed actuator can operate stably along a 20 mm guider. The proposed design is suitable for precision motion control applications.

## 1. Introduction

The wafer probe station is the key testing equipment for electrical performance measurement in the manufacturing process of integrated circuits [1,2,3]. To realize high efficient tests of wafers with hundreds of dies, the motion servo system needs to be fast, stable, and precise within a long travel range. As the die size on the wafer decreases and the wafer diameter increases, the number of dies on a wafer increases significantly. Consequently, the demand for high speed, long travel stroke, and high-resolution motion servo systems becomes urgent. Due to characteristics like a fast response, high resolution, and long stroke, piezoelectric actuators [4,5,6,7] have been applied in many areas, such as aeronautics and precision instrument [8,9,10]. This work provides a potential solution of piezoelectric motors for the wafer probe station.

After several decades of development, many piezoelectric actuators with different principles have been proposed. According to vibration state, they can be classified into resonance piezoelectric actuators and non-resonance piezoelectric actuators. The resonance actuators are fast and compact [11,12,13,14]. However, the control strategy of resonance systems is complex since both the mechanical actuator and its power source exhibits high nonlinearity [10,15]. Consequently, a motion system actuated by resonance actuators always needs closed-loop control [16,17,18,19,20]. The non-resonance piezoelectric actuators avoid high-frequency resonance, so it is suitable for both open-loop and closed-loop motion systems.

Among non-resonance piezoelectric actuators, there are two main classes, inertial actuators and inchworm actuators [21]. The inertial actuators are pushed by the reaction force generated from an unsymmetrical mass vibration system, which is formed by either an unsymmetrical structure or voltage excitation [22,23,24,25]. These actuators have a simple structure and compact size. However, the reaction force is proportional to the weight of vibrating mass, so this type of actuator is suitable for small dimension applications.

Inchworm actuators have high thrust and high resolution [26,27,28], and utilizing the inchworm principle increases the stroke by accumulating paces of the actuator and can obtain very high resolution within one pace [29,30]. In the existing research on inchworm actuators [31,32], the way to realize clamping and release mainly depends on the deformation of laminated piezoelectric ceramics or the deformation of mechanical amplification mechanism actuated by laminated piezoelectric ceramics. The deformation output of laminated piezoelectric ceramics is micron level, therefore, all the above mechanisms have high requirements for structural machining accuracy. After running for some time, with the wear of the clamping mechanism, the performance of the actuator will decline rapidly and fail. Moreover, the output force of the actuator during operation is quite different from the locking force after power failure. Therefore, these designs demonstrate common disadvantages like short stroke and short life span. To overcome these drawbacks, Marth and Waldbronn [33] invented a piezoelectric actuator that is pushed by eight groups of piezoelectric legs. DSM [34] developed another design to realize long stroke by a specific linkage mechanism between two camping feet. Its motor is capable of a 20 mm stroke and speed of 12 mm/s. This design also adopted a differential clamping mechanism. These two designs show that it is possible to enlarge the stroke of the inchworm mechanism by differential clamping of two pairs of clamping feet.

In this research, an advanced inchworm piezoelectric actuator is proposed, whose major clamping feet are linked with a pair of auxiliary clamping feet by a spring, and one of the guiders is designed to be movable. This actuator adopts the spring servo differential clamping mechanism to realize effective disengagement and locking, which not only reduces the harsh requirements of the motor on machining accuracy, but also greatly improves the stability of the motor in the operation process and overcomes the performance degradation caused by wear in the existing mechanism. This method improves the service life of inchworm piezoelectric actuators. This work presents another long-stroke design of an inchworm piezoelectric walking actuator to obtain stable and precise linear motion. Construction and the principle of operation are introduced. Structure details were proposed and experimental tests of the prototype were conducted.

## 2. Principle

As shown in Figure 1, the morphology of the actual working surface of the guide mechanism has changed slightly, which makes the clamping mechanism not only unable to achieve a differential clamping state in actual work, but also has a significant impact on the clamping force. Therefore, it is difficult to produce alternating clamping in two groups of clamping mechanisms with a certain span by using the micron deformation of laminated piezoelectric ceramics. On the one hand, the guide mechanism needs to have a high degree of parallelism; on the other hand, the overall dimension of the clamping mechanism needs to be highly consistent.

To reduce the adverse effects caused by micromorphology deviation, a new inchworm motor with springs servo differential clamping mechanism is proposed in this paper. Its schematic diagram is shown in Figure 2. The actuator is composed of four groups of clamping mechanisms, a group of actuating modules, and a guiding mechanism. The clamping mechanisms are in two groups and are combined in an elastic connection. One clamping mechanism in each group is fixedly connected with both ends of the actuating module along the actuating direction. Here, the clamping mechanism directly and fixedly connected with the actuating module is called the clamping foot, while the clamping mechanism elastically connected with the main clamping is called the auxiliary clamping foot.

The components in the motor discussed here are similar to that of the traditional inchworm piezoelectric motor; that is, two clamping elements and one push element linking these two elements. While it is working, the push element stretches and elongates along with the alternating clamping of two clamping elements. The differences between the advanced one investigated in this work and the traditional inchworm motor lie in the self-adapting guider and two auxiliary clamping elements linked to major ones by springs, as shown in Figure 3.

Three voltage signals with different phases are connected to piezoelectric parts (11, 12, 3, 21, and 22 shown in Figure 3) to excite the needed deformation of each piezoelectric part. These three voltage signals are shown in Figure 4.

According to Figure 3 and Figure 4, the states of piezoelectric parts during one period are shown in Figure 5. The deformation phase of elements 11 and 12, and 21 and 22 are designed to be reversed, which assures the clamping state of 12 and 21.

There are six different states while the motor is working. In state 1, the major clamping element 12 and the auxiliary clamping element 22 elongate and clamp the guiders. In state 2, the pushing element 3 elongates and pushes the major clamping element one pace ΔX forward. In the meantime, the linking spring between 21 and 22 is compressed. In state 3, 11 and 21 elongate and clamp the guiders while the other elements stay at states the same as is observed in state 2. In state 4, 12 and 22 release the guiders, and 22 moves one pace forward due to the compressed spring. In state 5, 3 shrinks and 12 is pulled one pace forward as the linking spring between 11 and 12 is stretched. In state 6, 12 and 22 elongate again and clamp the guiders. While 11 and 21 release the guider, auxiliary clamping element 11 is pulled one pace forward by the stretched spring. As a result, the motor returns to state 1 and if the exciting voltage signals are applied periodically, the motor keeps step movement continuously.

According to the analysis above, the motor moves ΔX during one period, so the speed of this motor is formulated as below.
*V* = ΔX*f*,(1)
where *f* is the frequency of driving voltage and *V* is the speed of this motor. Provided that there is no slip between clamping elements and guiders, the thrust of this motor can be expressed by
*F* = 2 *μ_s_F*_0_ − *k_l_*ΔX,(2)
where *F*_0_ is preload applied on the self-adapting guider, *F* is the thrust of this motor, *k_l_* is the stiffness of linking springs, and *μ_s_* is static friction constant.

When designing this motor, the following principles must be fulfilled.
*K_p_
*>> *k_l_*, *k_p_* >> *k*_0_,(3)
where *k_p_* is the stiffness of clamping elements, *k_l_* is the stiffness of linking springs, and *k*_0_ is the stiffness of preloaded spring. In this way, the clamping force is maintained while the motor travels along the guiders and avoids the potential decrease in performance raised by the wear of clamping elements. Additionally, in the actual design, the location of major clamping elements and auxiliary clamping elements shall be very near to each other so this motor is not sensitive to assembling errors which cause difficulty of clamping.

## 3. Prototype Investigation

### 3.1. Structure Design

According to the principle discussed above, an advanced inchworm linear piezoelectric motor was proposed, as shown in Figure 6. Since this prototype is designed to verify the principle, all parts of the motor were made of steel. The multilayer piezoelectric ceramics are products of PI Company, PL055.30 (Physik Instrumente (PI) GmbH & Co. KG, Karlsruhe, Germany), whose performances are listed in Table 1. The structure of the actuator is mainly designed according to the size of piezoelectric ceramics and the required preload. In addition, the natural frequency of the structure also needs to be much greater than the working frequency of the actuator.

In Figure 6, all clamping elements are adopted level structures that amplify the deformations of multilayer piezoelectric ceramics. All multilayer piezoelectric ceramics are preloaded by screws or wedges, and the whole body of the motor is machined at one time, which ensures the position precision between clamping elements.

In Figure 7 and Figure 8, details of the pushing element and clamping element are shown. The flexible hinges act as the pivot point of the lever mechanism and they also provide preload for multilayer piezoelectric ceramics. Surfaces for supporting multilayer piezoelectric ceramics must be strictly parallel and adjustable supporting pads are used to ensure multilayer piezoelectric ceramics suffer even pressure. Moreover, the contacting surfaces of major clamping elements and auxiliary clamping elements are close to each other so that these surfaces could clamp the guiders alternatively regardless of the processing errors. The pushing element in Figure 6 uses one piece of PL055.30 and a guiding mechanism is utilized to provide preload and link two major clamping elements. The wedge mechanism is utilized to apply the needed preload.

According to the above analysis, a prototype was made, as shown in Figure 9. The length of the guide rail is 20 mm. To verify the feasibility of the principle of the actuator, the driving feet of the prototype are in steel-to-steel contact with the mover, and the contact surface is ground.

### 3.2. Experimental Verification

The experiment system of the prototype is shown in Figure 10. The system consists of an oscilloscope, two power supplies, a drive circuit, a laser displacement sensor, and the motor prototype. The experiment system was set on the vibration isolation platform. The model of laser displacement sensor is LK-G series CCD laser displacement sensor of KEYENCE company. All experiments are carried out on the actuator under the open-loop control.

After the prototype is assembled, the first thing is to confirm the contact states between the motor and guiders. Table 2 lists the displacement responses of multilayer piezoelectric ceramics in pushing element and clamping elements. The data in the second column are the displacement responses of each element before the motor is assembled into guiders and in the third column are the displacement responses of clamping elements after they are preloaded by guiders. Due to the limitation of the mechanical structure of the prototype, the displacement responses of the piezoelectric part 3 after reloading cannot be measured by the laser displacement sensor. These data, when multilayer piezoelectric ceramics are applied with 100 V rectangular voltage, are obtained by CCD laser displacement meter LK-G series of KEYENCE Company (KEYENCE Corporation, Osaka, Japan), which has a resolution of 50 nm and the sampling time is 200 μs. Although the responses are not the same, the key to realizing this principle is that the adjacent clamping elements alternatively contact and clamp the guiders. So the prototype fulfills this requirement.

After the experimental system is built, the actuation characteristics of the prototype under no-load are tested. The step distance of the actuator is small and the working frequency is low, therefore, in order to avoid the resonance of the actuator structure caused by too high working frequency, a lower working frequency lower than 50 Hz is selected here. The pre-pressure of 10 N was applied to the movable guide rail. Under the excitation of three groups of square wave voltage signals with the same frequency and two with the same phase difference, the operating frequency of the actuator was changed, and the operating curve of the actuator was measured.

As shown in Figure 11, it is the travel curve of the prototype under the excitation frequency of 30 Hz. Figure 11a shows the travel curve of the prototype within 13 s. Due to the surface defect of the contact between the guide rail and the clamping feet, the speed of the prototype changes slightly during the travel. Figure 11a shows that the travel curve is not straight, and the prototype travels for 250 μm within 13 s. The average speed is 19 μm/s; Figure 11b shows the walking curve of the prototype within 1 s. It can be seen that the prototype has advanced 30 steps within 1 s, and the distance is about 19 μm. Therefore, the step distance of the prototype is about 0.63 μm. It can be seen from the curve of each step that there is a certain reciprocating oscillation in each step of the prototype, which may be caused by the overshoot of the square wave signal when charging the capacitive load laminated piezoelectric ceramics.

Figure 12 shows the travel curve of the prototype under the excitation frequency of 22 Hz. Figure 12a shows the travel curve of the prototype within 13 s, and the prototype travels for 156 μm within 13 s. The average speed is 12 μm/s; Figure 11b shows the walking curve of the prototype within 1 s. It can be seen that the prototype has advanced 23 steps within 1 s, and the distance is about 12 μm. Therefore, the step distance of the prototype is about 0.52 μm. It can be seen from the data analysis of Figure 11 and Figure 12, that there is a small deviation in the step distance of the prototype under different frequencies, which may be caused by the sliding between the clamping foot and the contact surface under high-frequency driving.

During the no-load operation of the prototype, the characteristics of the excitation signal was kept unchanged, except the excitation frequency. The relationship curve between the output speed and the excitation frequency is shown in Figure 13. It can be seen that when the excitation frequency of the prototype is lower than 24 Hz, its operation speed increases linearly with the excitation frequency. When the excitation frequency is further increased, its operation speed increases faster. This is because when the operating frequency of the prototype increases, the running speed of the prototype increases, and the sliding between the clamping feet and the guide rail increases, so that the speed increase becomes larger.

This experiment is conducted under different driving frequencies. As can be seen from Figure 13, the speed increases along with the driving frequency, which verifies Equation (1). Experiments show that the prototype can stably realize the two-way stepping movement with a large stroke, which is easier to realize than the actuator based on the traditional inchworm principle. In fact, when we fix the movable guide rail on the base with the clamp without the auxiliary clamping feet, it is difficult for the actuator to run continuously, which also proves that the above differential clamping mechanism can indeed improve the working stability and stroke of the actuator, and reduce the influence of machining error on the actuator. Here, the preload is not further increased because the clamping feet adopt a displacement amplification mechanism, which reduces its stiffness. In addition, although the surfaces of the actuator clamping feet and the guide rail have been ground, the surfaces have not been hardened, and the deformation of the clamping feet is in the same order of magnitude as the fluctuation range of the microscopic appearance of the contact surface. This makes the contact between the clamping feet and the guide rail difficult to disengage under a large preload, so it can not work normally.

## 4. Conclusions

To overcome problems that arose by wear and machining errors in inchworm piezoelectric actuators, two auxiliary clamping elements are linked to major clamping elements by springs, and one of two guiders is designed to be adjustable and preloaded by a spring. The operation procedure of the advanced inchworm piezoelectric actuator was analyzed by demonstrating each state of the actuator in one working period. The factors that determine the performances of this actuator were investigated, as well as the requirements for stiffness of every elastic part in the actuator. Prototype testing verified the principle and a method to examine the assembling state of the actuator was also presented.

In summary, the introduction of auxiliary clamping feet makes the above actuator have the following advantages over the traditional design:The operation stability of the actuator is improved. The actuator has the same clamping force whether it is powered on or off. Even if the parallelism of the guide rails on both sides is not high, the differential clamping feet close to each other can still operate normally;The cost of the actuator is reduced. Laminated piezoelectric ceramic is the core component of the actuator. Due to the use of differential clamping, the clamping can be realized by using laminated piezoelectric ceramic with a small volume;The actuating stroke of the actuator is increased. With the traditional design, it is often difficult to realize the action of large stroke because the parallelism of the guide mechanism is not high. The above design reduces the requirements for the parallelism of the guide rail and can realize stable operation in a large stroke range.

In future research work, we will further improve and optimize the prototype, study the influence of different friction loads (such as Alumina/Alumina contact) on the performance of the actuator, and test the thrust of the actuator.

## Figures and Tables

**Figure 1 micromachines-13-00412-f001:**
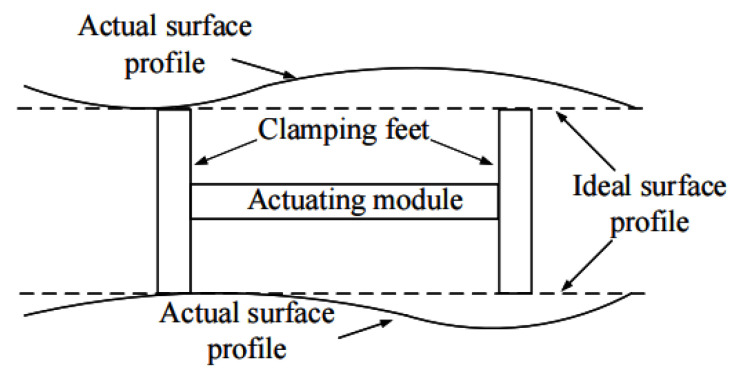
Influence of surface morphology of guide rail on motor clamping.

**Figure 2 micromachines-13-00412-f002:**
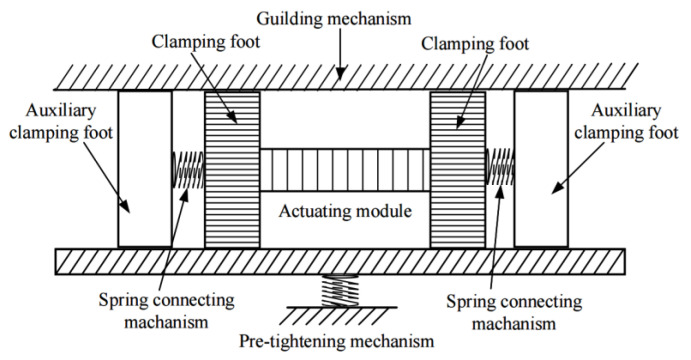
Structural diagram of auxiliary clamping feet inchworm piezoelectric linear motor.

**Figure 3 micromachines-13-00412-f003:**
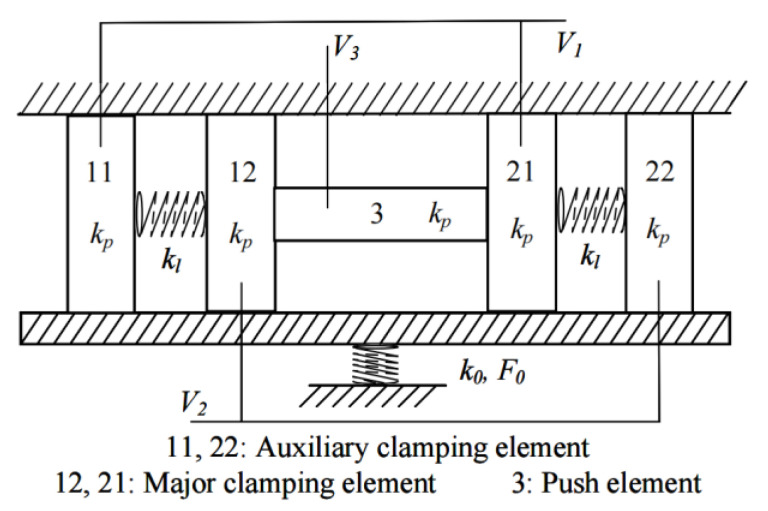
Sketch view of advanced inchworm piezoelectric motor.

**Figure 4 micromachines-13-00412-f004:**
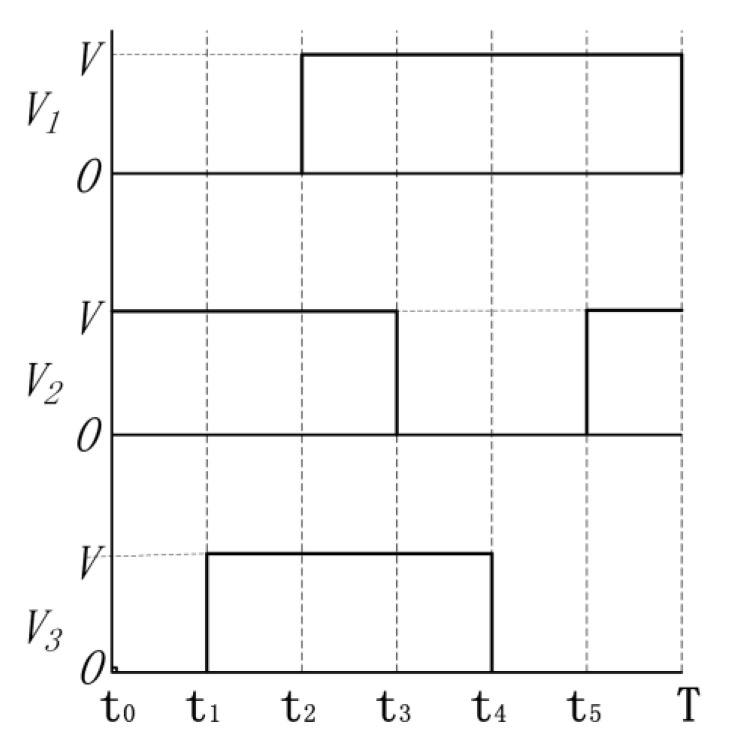
Exciting voltage signals applied to piezoelectric elements.

**Figure 5 micromachines-13-00412-f005:**
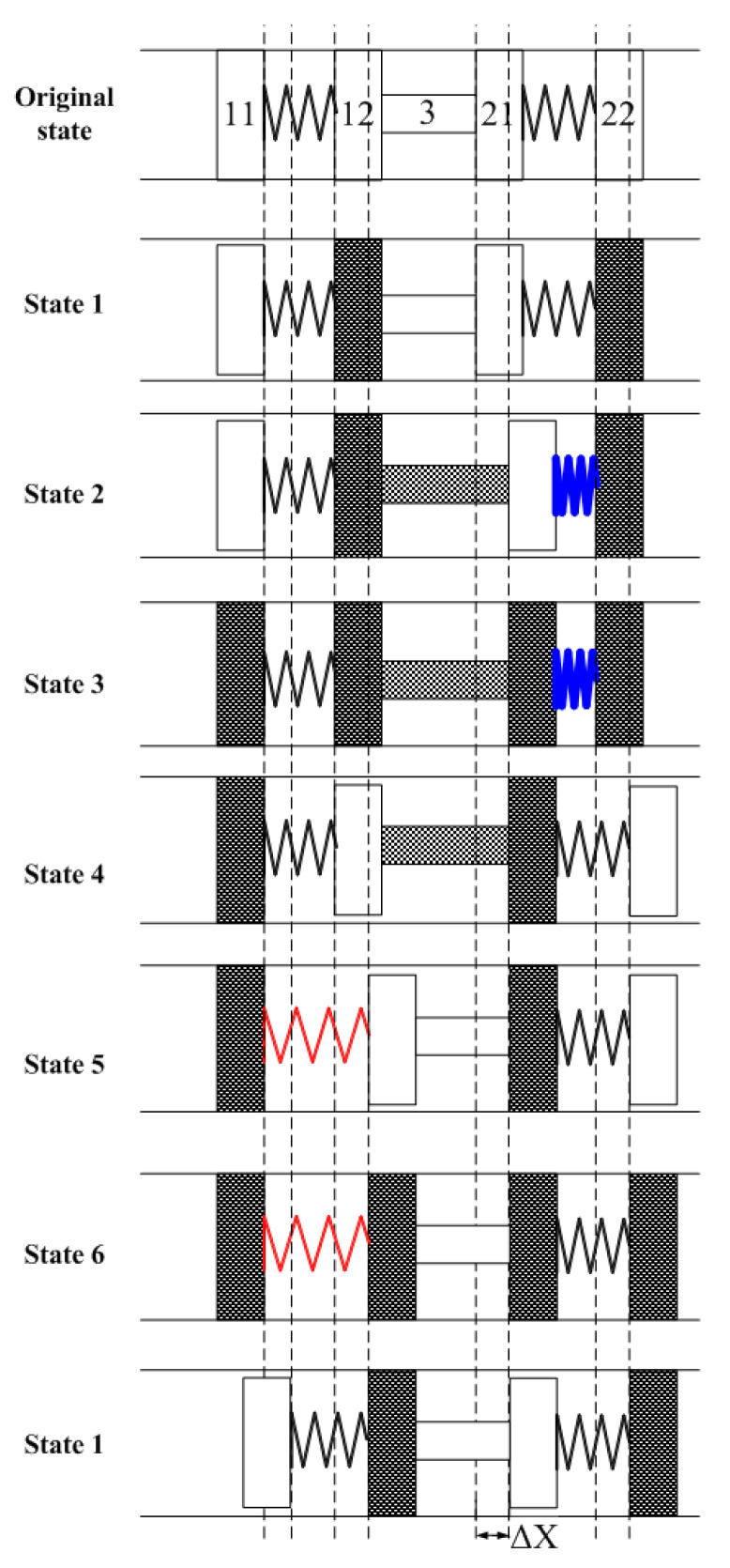
States of the actuator during one operation period.

**Figure 6 micromachines-13-00412-f006:**
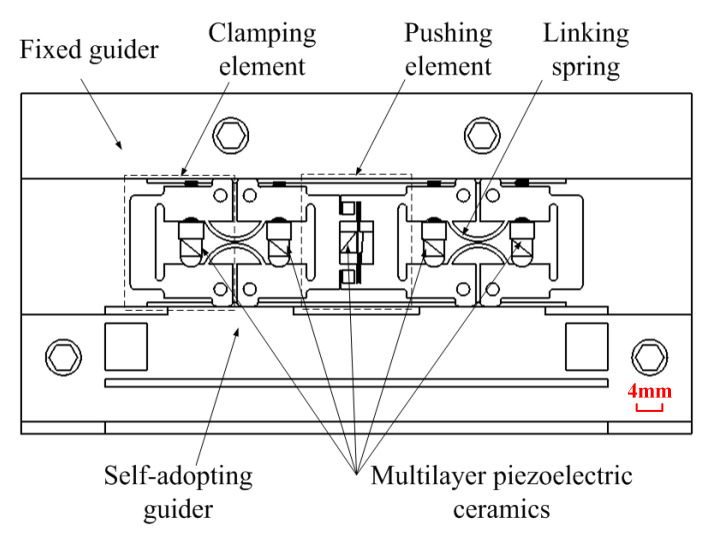
Structure of advanced inchworm piezoelectric motor.

**Figure 7 micromachines-13-00412-f007:**
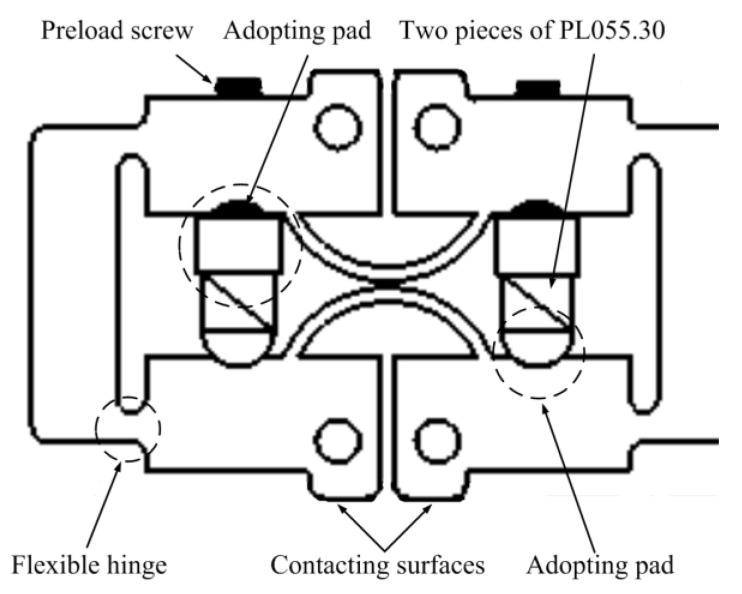
Details of clamping elements.

**Figure 8 micromachines-13-00412-f008:**
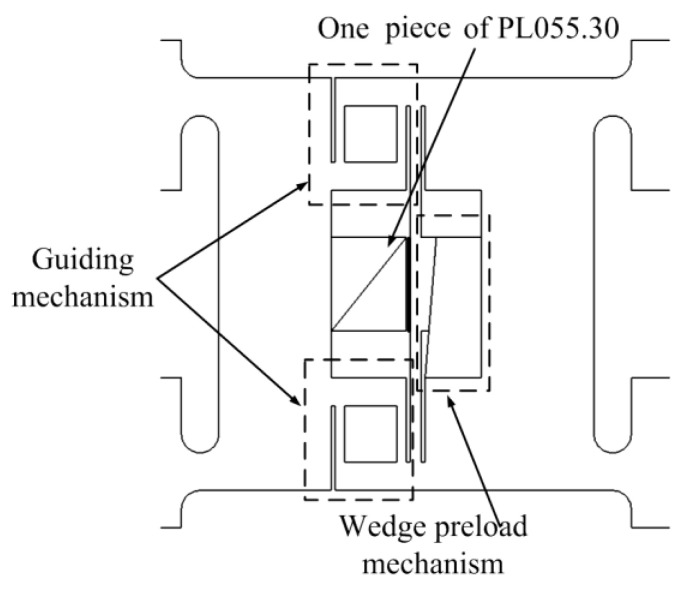
Details of pushing element.

**Figure 9 micromachines-13-00412-f009:**
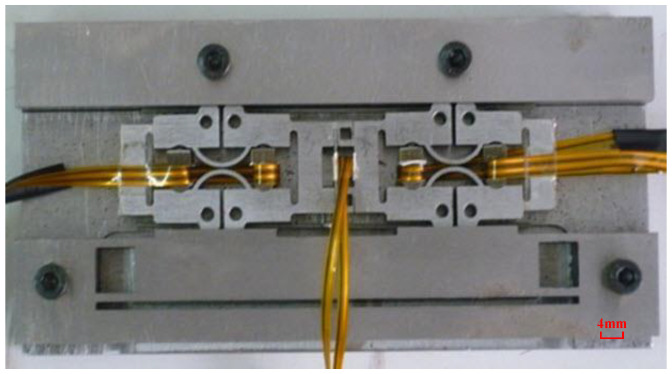
Picture of the prototype.

**Figure 10 micromachines-13-00412-f010:**
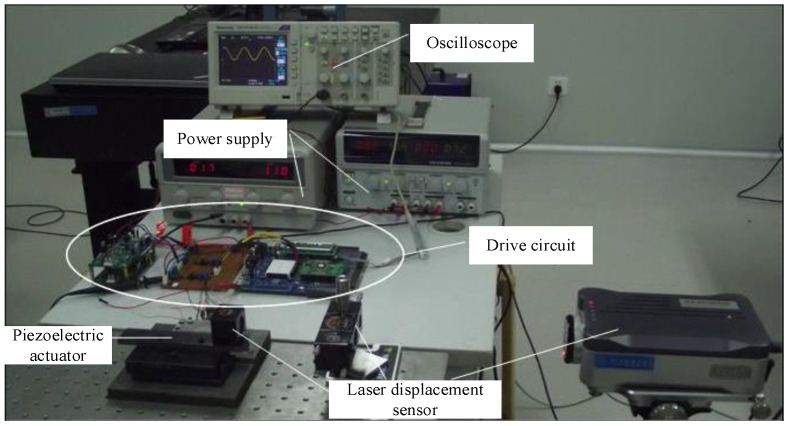
Experiment system.

**Figure 11 micromachines-13-00412-f011:**
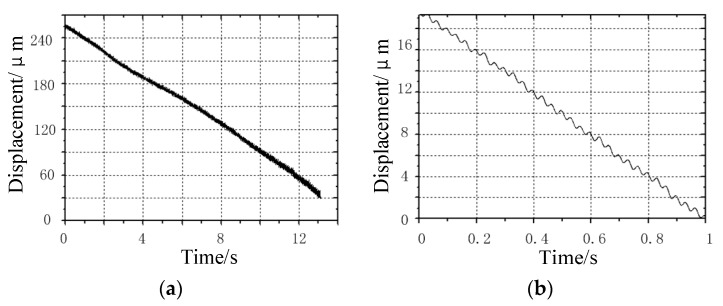
No-load operation curve of the prototype under the excitation frequency of 30 Hz: (**a**) travel curve of the prototype within 13 s; and (**b**) Travel curve of prototype within 1 s.

**Figure 12 micromachines-13-00412-f012:**
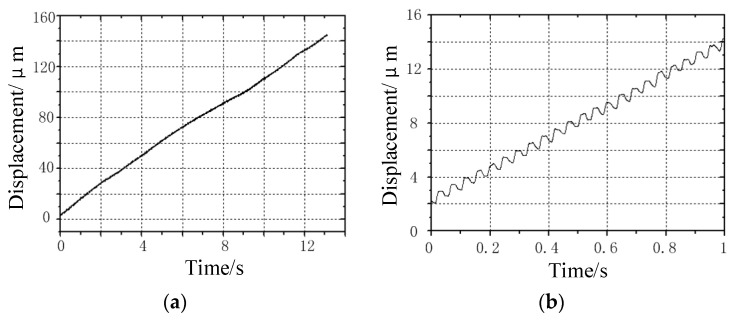
No-load operation curve of the prototype under the excitation frequency of 22 Hz: (**a**) travel curve of the prototype within 13 s; and (**b**) Travel curve of prototype within 1 s.

**Figure 13 micromachines-13-00412-f013:**
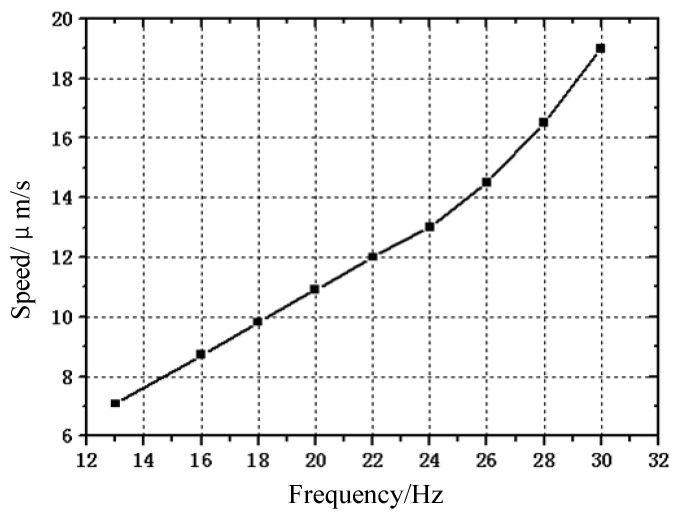
Moving speed versus driving frequency.

**Table 1 micromachines-13-00412-t001:** Performance of multilayer piezoelectric ceramics PL055.30.

Dimensions (mm)	Nominal Displacement (μm@100 V) ± 20%	Blocking Force (N)	Electrical Capacitance (nF) ± 20%	Resonance Frequency (kHz)
5 × 5 × 2	2.2	>500	250	>300

**Table 2 micromachines-13-00412-t002:** Displacement response of motor at 100 V (μm).

Element	Before Assembly	After Assembly
11	7.3	3.9
12	7.6	5
3	3.8	-
21	5.7	3.8
22	6.7	4.6

## Data Availability

Not applicable.

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
