# Peer review of "Long Stroke Design of Piezoelectric Walking Actuator for Wafer Probe Station"

_micromachines, 2022, doi:10.3390/mi13030412_

Round 1

Reviewer 1 Report

Dear Authors,

The paper needs major revision. There are plenty of questions and comments listed in the attachment.

Kind Regards

Reviewer 2 Report

COMMENTS TO THE AUTHORS

This paper presents and tests a long stroke design of piezoelectric walking actuator. The authors claim that the main contribution of this work is that the proposed actuator can achieve long stroke. However, as far as I am concerned, the quality of the paper is not enough to be published, the reasons are as follows:

  1. In the introduction part, Page 2, Line 46, “these design own common disadvantages like short stroke and short life span”. Is there any relevant literature support this point? It is misleading for me, piezoelectric walking actuator can achieve long stroke and long-life span, it can move step by step to accumulate a long stroke, and long-life span can be achieved because it operates in low frequency and is usually driven by static friction force.

  1. What is the main contribution of this work? The long stroke can be achieved by many previous piezoelectric walking actuators.

  1. What is the design principle? The dimension and size of the elements are not provided.

  1. The experimental results are insufficient, how about the output force, the resolution, the repeatability of the step?

  1. There are some grammar errors, the authors should check the manuscript clearly.

In page 1 Line 22, “As the size of dies on wafer is get smaller”

The author should be clear whether there is a space between numbers and units.

The style of the figures is not uniform.

Round 2

Reviewer 1 Report

Dear Authors,

My questions and comments have been adressed in the revised version of the paper. Still there are some minor mistakes, spelling mistakes. Please pay attention to rules of writing scientific papers. Please check references again for the right forms like the one below. 

Reference 20 should be written as follows;

[20]  Koc, B.; Delibas, B. Method for Closed-Loop Motion Control for an Ultrasonic Motor. US Patent 16/633,940, 25 June 2020.

Kind Regards

Author Response

Thank you for your professional and valuable questions. We have carefully revised the manuscript and checked the format of references.

Reviewer 2 Report

Although the authors have replied to most of my comments, It is my opinion that the developed actuator does not have many advantages in mechanical characteristics.

Author Response

Thank you for your valuable and professional opinions. The prototype in this paper is the first generation prototype, and there are some deficiencies in performance. In the future, we will further optimize the motor in terms of material and mechanical structure, so as to further improve the motor performance.